# Evaluation of the Performance of Rapid Diagnostic Tests for Malaria Diagnosis and Mapping of Different *Plasmodium* Species in Mali

**DOI:** 10.3390/ijerph21020228

**Published:** 2024-02-15

**Authors:** Pascal Dembélé, Mady Cissoko, Adama Zan Diarra, Lassana Doumbia, Aïssata Koné, Mahamadou H. Magassa, Maissane Mehadji, Mahamadou A. Thera, Stéphane Ranque

**Affiliations:** 1Institut Hospitalo-Universitaire Méditerranée Infection (IHU), Aix Marseille Université, 13005 Marseille, France; dembelepascal@ymail.com (P.D.); adamazandiarra@gmail.com (A.Z.D.); dolslassina@yahoo.fr (L.D.); maissane.mehadji@gmail.com (M.M.); 2Aix-Marseille University, IRD, AP-HM, SSA, VITROME, 13005 Marseille, France; 3Programme National de Lutte Contre le Paludisme (PNLP), Bamako BP 233, Mali; madycissoko@ymail.com (M.C.); mahamadouhmagassa@yahoo.fr (M.H.M.); 4Malaria Research and Training Center (MRTC), FMOS-FAPH, Mali-NIAID-ICER, Université des Sciences, des Techniques et des Technologies de Bamako, Bamako BP 1805, Mali; mthera@icermali.org; 5Laboratoire de Biologie Moléculaire Appliquée (LBMA), Université des Sciences, des Techniques et des Technologies de Bamako, Badalabougou, Bamako BP 423, Mali; aissatakone700@gmail.com

**Keywords:** *Plasmodium* species, Malaria, rapid diagnostic tests, sensitivity, specificity, PCR, Mali

## Abstract

Background: The first-line diagnosis of malaria in Mali is based on the use of rapid diagnostic tests (RDT) that detect the Histidin Rich Protein 2 (HRP2) antigen specific to *Plasmodium falciparum*. Our study, based on a real-time polymerase chain reaction (qPCR) gold standard, aimed to describe the distribution of the *Plasmodium* species in each administrative region of Mali and to assess the performance of RDTs. Methods: We randomly selected 150 malaria-negative and up to 30 malaria-positive RDTs in 41 sites distributed in 9 regions of Mali. DNA extracted from the RDT nitrocellulose strip was assayed with a pan-*Plasmodium* qPCR. Positive samples were then analyzed with *P. falciparum*-, *P. malariae*-, *P. vivax*-, or *P. ovale*-specific qPCRs. Results: Of the 1496 RDTs, 258 (18.6%) were positive for *Plasmodium* spp., of which 96.9% were *P. falciparum*. The *P. vivax* prevalence reached 21.1% in the north. RDT displayed acceptable diagnostic indices; the lower CI95% bounds of Youden indices were all ≥0.50, except in the north (Youden index 0.66 (95% CI [0.44–0.82]) and 0.63 (95% CI [0.33–0.83]. Conclusions: Overall, RDT diagnostic indices are adequate for the biological diagnosis of malaria in Mali. We recommend the use of RDTs detecting *P. vivax*-specific antigens in the north.

## 1. Introduction

Despite numerous efforts, malaria remains a major public health problem around the world [1]. According to the latest estimates of the World Health Organization published in December 2022, there were 247 million cases of malaria in 2021 versus 245 million in 2020 [2]. About 619,000 deaths were attributed to this disease in 2021 versus 625,000 deaths in 2020 [1].

The WHO African region alone will account for over 95% of all malaria cases and 96% of all deaths attributable to malaria in 2021. Four African countries account for almost half of all malaria cases worldwide: Nigeria (26.6%), the Democratic Republic of Congo (12.3%), Uganda (5.1%) and Mozambique (4.1%) [1]. Children under five years of age are the most vulnerable targets affected by malaria, constituting 80% of all malaria cases and 96% of all malaria deaths in 2021 [1]. Mali is one of the countries with the highest number of cases and associated deaths. The malaria case fatality was estimated to be 1.25‰ or 1498 deaths out of 1,197,864 confirmed severe cases. However, malaria remains a major public health concern as it represents one of the main reasons for medical visit in public health facilities with a morbidity rate of 34% and mortality rate of 22% [3].

Children under 5 and pregnant women are the most affected [3]. The main species encountered were *Plasmodium falciparum* (over 85%), *Plasmodium malariae* (10–15%) and *Plasmodium ovale* (1%) [4]. *P. falciparum* causes severe and complicated forms of malaria and has a high fatality rate [4]. Notably, cases of *Plasmodium vivax* have been documented in Mali [5,6]. 

In 2018, the prevalence of malaria in children aged 6–59 months was 19% according to RDT results. The highest prevalences were found in the regions of Sikasso, Segou, Mopti, and Koulikoro (30%, 26%, 25% and 22%, respectively). The prevalence was low, between 1 and 10% in the regions of Kidal, Taoudenit, and Timbuktu and <1% in the district of Bamako. In the north region, the malaria prevalence in Gao and Menaka was >10% [7]. The infant and child mortality was 101‰ [7]. In 2021, an RDT test-based Malaria Indicator Survey in Mali (MIS 2021) reported a stable 19.4% national prevalence among children aged 6–59 months [8]. Despite the great efforts made by the State and its partners, the 125‰ malaria incidence remains below the expected target of 81‰ in Mali following the DHIS2 2020 (District Health Information Software) [3].

The national malaria control policy in Mali is in line with several international initiatives [9] aiming to “Roll Back Malaria” [10]. Case management through early diagnosis and effective treatment remain the cornerstone of the fight against this disease. 

To diagnose suspected cases of malaria, Mali uses microscopic examination of thick and thin blood smears and/or RDTs based on lateral flow immunochromatography [11]. The introduction of RDTs as a diagnostic tool for malaria between 2005 and 2009 has greatly improved the quality of malaria case management in Mali. The proportion of suspected cases who received a biological diagnosis by RDT was 87% and RDTs remain the most widely used tool in health facilities [12].

The RDT is based on the detection of the antigen Histidin Rich Protein 2 (HRP2) antigen in the blood, specific to *Plasmodium falciparum*, which is the most prevalent species in Mali [11]. It is most frequently used in primary healthcare facilities and at the community level by community health workers (CHWs) because of its easy handling [13,14]. It is advantageous in providing results within 15 min and discriminating malaria from non-malaria fevers through the detection of at least one specific antigen. The most used antibodies react to HRP2; other antigens, namely aldolase and plasmodial lactate dehydrogenase (pLDH), are less frequently used. Proper use of RDTs should optimize malaria diagnosis and avoid antimalarial drug resistance; however, the increase in false-negative RDT results poses a new challenge to malaria control [15,16]. Malaria RDT false-negative results can be explained by asymptomatic malaria with parasite densities below the RDTs’ limit of detection [17,18]. Another explanation is the increasingly reported deletion of the *PfHRP2/3* genes encoding the HRP2 protein. Whatever the cause, false-negative RDT results in untreated patients who carry parasites, which maintain malaria transmission [19]. All of these issues are a serious threat to National Malaria Control Program (NMCP) [20]. PCR-based diagnostic tests display improved sensitivity, specificity for *Plasmodium* species detection and identification. PCRs detect mixed infections more accurately than traditional methods [21,22]. Although the PCR is not currently optimized for routine diagnosis, it is of major importance in epidemiological studies [23].

However, studies conducted in northern and central Mali have documented other species of *Plasmodium* such as *P. malariae*, *P. vivax*, and *P. ovale* with a prevalence ranging from 1 to 20% [5,6]. In the context of strengthened malaria control policies and the emergence of *Plasmodium* species other than *Plasmodium falciparum*, the present study aimed to provide useful information on (i) the distribution of different *Plasmodium* species in Mali and (ii) the performance of rapid diagnostic tests compared with qPCR.

## 2. Methods

### 2.1. Study Sites

Mali is a landlocked country in West Africa located between the 10- and 25-degree north latitudes and between the 4- and 12-degree west longitudes. It covers a 1,241,238 km^2^ area. Malaria is generally endemic in most parts of Mali, with a recrudescence during the rainy season [24]. The level of malaria endemicity in Mali varies from one eco-climatic region to the other. Factors responsible for variations in endemicity include rainfall, altitude, temperature, hydro-agricultural development and urbanization [24]. There are 3 climatic zones in Mali that extend from south to north: the Sudano-Guinean zone, which covers 25% of the territory and has a rainfall of approximately 1300 to 1500 mm per year; the Sahelian zone, which covers 50% of the territory and receives rainfall of 200 to 800 mm per year; and the Saharan desert zone, which represents 25% of the territory. This zone is marked by irregular rainfall, often less than 200 mm per year. For this study, we collected RDT cassettes from 41 sites in the following regions of Mali: Kayes, Koulikoro, Sikasso, Ségou, Mopti, Timbuktu, Gao, Kidal and Menaka, from June to December 2021. No sample was collected in the Taoudeni region or the capital city Bamako.

### 2.2. Study Design

#### 2.2.1. Rapid Diagnostic Test (RDT)

In Mali, the purchase and distribution of rapid diagnostic tests is an essential part of the master plan for the supply and distribution of essential medicines (SDADME). To control consumption needs and guarantee their availability and quality, they are managed similarly to other medicines, by using the same management tools. Decision No. 2011-774/MS-SG of 11 July 2011 made their application mandatory. Its objective is to ensure the correct supply of health products to the population by the Popular Pharmacy of Mali (PPM), which is the State’s preferred tool for the supply, storage and distribution of health products through a State-PPM contract plan. This system is supplemented by the private sector through the Private Import and Wholesale Establishments (EPIWG) of approved private suppliers of pharmaceutical products. The RDTs used in our study sites were the SD Bioline Pf Ag (Standard Diagnostics, Inc, 05FK50), Adv Dx Malaria Pf Ag HRP2 (J. Mitra & Co. Pvt. Ltd, IR016025), which both detect the *P. falciparum* specific Histidin-rich Protein 2; and First Response Malaria Ag pLDH/HRP2 (Premier Medical Corporation Ltd, I16FRC25), which detects the protein Lactate Dehydrogenase (pLDH) of *P. falciparum* and screens for other *Plasmodium* species (Pan). RDTs were performed by health workers to reach a diagnosis of malaria in febrile outpatients presenting at health facilities (Figure 1). They were selected proportionally to the different collection sites in the region before including positive and negative RDTs.

#### 2.2.2. Inclusion and Exclusion Criteria

Our study included RDT cassettes that were either negative for *P. falciparum*, with a single band visible in the control window “C” Test, or positive for *P. falciparum*, with a band both in the “C” control and in the “T” test windows. RDT cassettes with no band in the “C” control window were excluded.

#### 2.2.3. DNA Extraction Technique

The selected RDT cassettes were opened under a Type 2 Microbiological Safety Station (MSS) using a dissecting needle (LANCEOLEE Models LT2304, Pakistan), dissecting forceps, and a pair of scissors. The nitrocellulose tape was removed from the plastic cassette and then cut into small 3 × 3 mm pieces using scissors for each sample. The scissors were decontaminated between each sample with 70° ethanol [25]. The cut nitrocellulose samples were then introduced into 1.5mL collection tubes for incubation at room temperature in 400 µL of Nuclisens EasyMag Lysis Buffer (bioMérieux, Craponne, France). The tubes were then centrifuged for 2 min at 13,000 rpm using Lyse&Spin basket tubes (Qiagen, Courtaboeuf, France) to retain the nitrocellulose pieces and carry out extraction of the filtrate. DNA was extracted from 200 µL of the filtrate following the DNA Blood EZI Advanced XL protocol (QIAGEN Instruments Hombrechtikon, Switzerland), with a final elution volume of 50 µL. Random samples were assayed using a Qubit fluorometer (QUBIT2.0, Life Technologies Villebon sur Yvette, France) to control the DNA extraction quality. The DNA was then stored at −20 °C.

#### 2.2.4. Plasmodium Species Detection by qPCR

DNA extracted from the nitrocellulose strip of malaria-positive and malaria-negative RDT cassettes were first subjected to pan-*Plasmodium* qPCR. In a second step, only the samples that were positive were tested with qPCR systems specific for *P. falciparum*, *P. malariae*, *P. vivax,* or *P. ovale* to identify the *Plasmodium* species [26]. For the amplification reactions, we used 100 μL of Roche Mix (Roche diagnostics GmbH, Mannheim, Germany), 18 μL of each primer (forward and reverse), 12 μL of probe (Var ATS Probe) and 2 μL of distilled water. Then, 20 μL of this mix plus 2 μL of added DNA were distributed in the plate. For the first (pan-*Plasmodium*) qPCR, the samples were deposited in duplicate with three negative controls and one positive control. The PCR conditions are summarized in (Table 1). Real-time PCRs were performed on a CFX96 (BIO-RAD, Marnes-la-Coquette, France) or LightCycler480 II (384-well) (Roche Diagnostic International Ltd, Rotkreuz, Switzerland) thermocycler. The amplification reaction program for the identification of the plasmodial species was as follows: 2 min at 50 °C, 5 min at 95 °C, 45 cycles (10 s at 95 °C, 30 s at 54 °C, 1 min at 60 °C), and 30 s at 40 °C [27]. Several negative controls and a positive control for each species were included in each PCR plate. Samples with an amplification threshold (Ct) value less than 39 were considered positive.

#### 2.2.5. Case Definition

We considered the *Plasmodium*-specific PCR assay as the malaria diagnosis gold standard. Irrespective of the result, a malaria case had to display a positive *Plasmodium*-specific PCR result. True positive = (positive RDT/positive qPCR), false positive = (positive RDT/negative qPCR); false negative = (negative test/positive qPCR), true negative = (negative RDT/negative qPCR). Sensitivity was the proportion of positive RDT among the positive qPCR malaria cases. Specificity was the proportion of negative RDT among the negative qPCR non-malaria cases.

### 2.3. Data Analyses

Data were processed in Excel (version 2013) and analyzed using JavaStat 2-way contingency table analysis [28]. The diagnostic performance of malaria RDTs was determined via sensitivity (Se), specificity (Sp), and Youden and Kappa indices, each with their 95% confidence intervals (CI95%). The agreement between the two diagnostic methods was estimated using Cohen’s Kappa index [29]. The Kappa index indicates strong agreement when greater than 0.8 and poor agreement when less than 0.53. The Youden index (Y = Se + Sp − 1) combines both sensitivity and specificity to assess the validity of a diagnostic test.

### 2.4. Ethical Considerations

Our study was authorized by the National Malaria Control Program (number 042/MSDS-SG/NMCP, 4 February 2023), in accordance with Malian regulations on ethics and medical research.

## 3. Results

During the study period, 3338 malaria RDT cassettes were collected from 41 Health facilities in the 9 regions of Mali, of which 1019 (30.5%) were positive (Table 2). In Table 2, the number of RDTs and the prevalence of positive RDTs for malaria is detailed in each study site. Overall, 3338 RDT cassettes were collected, of which 69.5 tested malaria negative Appendix A.

We then randomly selected 30 positive and 150 negative RDTs in each administrative region for further qPCR analyses, except in two regions, namely Menaka and Kidal, where all positive tests (21 and 29 for Menaka and Kidal, respectively) and all negative tests (*n* = 36) in Menaka were included. Among the 1502 malaria RDT cassettes selected, 6 RDTs were excluded as invalid, 260 were positive and 1236 negative for malaria. All 1496 RDTs were further analyzed using *Plasmodium* spp. qPCR. Among them, 258 had a positive *Plasmodium* spp. qPCR. Among these 258 *Plasmodium*-positive samples, 251 (96.9%) were positive for *P. falciparum*, 5 (1.9%) for *P. vivax* and 2 (0.8%) for *P. malariae* (Figure 2). We detected two cases of mixed infections, where *P. falciparum* was combined either with *P. vivax* or *P. malariae*, in Kidal or Koulikoro, respectively (Table 3).

The *P. falciparum* distribution was ubiquitous with a prevalence ranging from 78.9% to 100% of the positive PCRs, depending on the region. *P. vivax* was identified in 21.1% (4/19) and 4.3% (1/23) of the positive PCRs in Menaka and Kidal, respectively. *P. malariae* was identified in 2.9% (1/34) and 3.4% (1/29) of the positive PCRs in Mopti and Koulikoro, respectively. *P. ovale* was detected in none of our samples. Remarkably, we identified two cases of *P. vivax*, for example, in Menaka (4) (Figure 3) and Kidal (1), on negative RDTs, which only detected the HPR2 antigen specific to *P. falciparum*.

We observed 77 discordant RDT results, namely 14.3% (37/258) false-positive and 3.0% (40/1236) false-negative results.

We observed both the high sensitivity (Se) (96.6%) and high specificity (Sp) (98.7%) of the RDTs in the Koulikoro region. In contrast, Se and Sp were lowest in the Kidal (73.9% and 92.4%, respectively) and Menaka (78.9% and 84.2%, respectively) regions. The Youden and Cohen’s Kappa indices of RDTs in each region are detailed in Table 4. Overall, RDTs displayed acceptable diagnostic indices. The lower bounds of the 95% confidence intervals (95%CI) of Cohen’s Youden and Kappa indices were all≥ 0.50, except in the Kidal and Menaka regions, where the Youden and Kappa indices were 0.66 (95% CI [0.44–0.82]) and 0.60 (95% CI [0.40–0.74], 0.63 (95% CI [0.33–0.83] and 0.62 (95% CI [0.32–0.80], respectively.

## 4. Discussion

Our findings confirm that *P. falciparum* is the most prevalent species overall in Mali. Yet, this study’s key finding is that non-*Plasmodium* species are also endemic, notably *P. vivax* in the northern region of Mali. Remarkably, we observed markedly lower diagnostic indices of malaria RDTs in these regions.

We report a *P. falciparum* prevalence ranging from 79% to 100% of the positive malaria RDTs according to the regions of Mali. Our results are consistent with those of Doumbo et al., who reported that *P. falciparum* accounted for 98.2% of the positive *Plasmodium* spp. tests since 1988 [4]. In 2011, an increase in the prevalence of *P. falciparum* from 74.13% during the dry season to 63.72% during the cold season was observed in a study by Koita et al. [6]. The 97% *P. falciparum* prevalence in the Mopti region is in line with Konate et al., who reported a 98% prevalence in Badiangara in 2020 [30]. This high, meso- to hyper-endemic, prevalence could be explained by the highly seasonal malaria transmission as well as the climate, which is characterized by 400 to 700 mm of rainfall per year, a short rainy season, from June/July to August/September, and a longer dry season. The region is irrigated by the Niger river and a tributary, the Yamé, which provides numerous breeding sites for *Anopheles gambiae* and *Anopheles funestus* during the 5 months of transmission each year [31]. 

Remarkably, we found a 21% and 4% prevalence of *P. vivax* in the Menaka and Kidal regions, respectively. Our results are in line with those of Bernabeu et al., who reported a 30% prevalence of *P. vivax* in five cities (Goundam, Timbuktu, Gao, Bourem, and Kidal) located in northern Mali [5]. Briefly, the *P. vivax* prevalence has been increasing in the northern part of the country, especially in Menaka, where we observed a higher prevalence than that observed by Koïta et al. in 2011.

In 2016, a study conducted in four West African countries, Burkina Faso, The Gambia, Ghana and Mali, verified that the prevalence of non-*falciparum* infections was higher in Mali (3.81%, 95% CI [2.22–5.68]) than in The Gambia (0.17%) [32]. To support this assertion, *P. vivax* and *P. malariae* have been found in black African women living in the Bamako area [32]. We also observed *P. malariae* infections in two regions of Southern Mali, namely 3.4% in Koulikoro and 2.9% in Mopti. Our findings are similar to those of a study carried out in the same Koulikoro region, which reported a 2.7% prevalence [33]. *P. malariae infection* is involved in nephrotic syndrome and chronic glomerulopathy, which can progress to end-stage renal failure [34,35] and has been associated with a high burden of anemia [36,37] and death [38]. It is therefore critical for the NMCP to control all non-*P. falciparum* human infections to achieve the malaria elimination goal.

We found no *P. ovale* in our samples. However, this parasite species, despite being less dangerous than *P. falciparum*, remains a public health burden that should be included in malaria elimination programs [39,40,41]. *P. ovale* has been reported to have a prevalence of around 2% in Mali. [42]. Conventional PCR followed by DNA sequence analysis is commonly used to differentiate the two subspecies *P. ovale curtisi* and *P. ovale wallikeri*, which have been documented in Mali and other West African countries [40,43,44]. 

The rapid detection and identification of *Plasmodium* parasite species with improved diagnostic tools are key for operative NMCPs [31]. In our study, the sensitivity of RDTs was higher than 80% in each region, except in Kidal and Segou, where it was moderate. The relatively lower performance of RDTs observed in these regions could be explained by the enhanced analytical sensitivity of the qPCR compared to RDTs. The majority of false-negative RDTs had either a fine band or weak staining, which could be related to low *Plasmodium* density in the blood, as already assumed in previous studies [45,46]. This condition could explain a decrease in both the sensitivity and specificity of RDTs. Another explanation could be the emergence of HRP-2/3 gene deletion [16,47]. The problem of false positives has been previously addressed; it could be due to the persistence of the HRP-2 protein in the patient’s blood up to several weeks after post-treatment parasite clearance [48]. In the Kidal and Menaka regions, the Youden and Kappa indices were lower than in the other regions. In Kidal, *P. vivax* was detected in a false-negative RDT that was designed to detect only the *P. falciparum*-specific HRP2 antigen. This relative emergence of non-*P. falciparum Plasmodium* species could be explained by the inability to detect other *Plasmodium* species of RDTs based solely on the detection of the *Pf HRP-2* antigen, which is specific to *P. falciparum*. Therefore, adapting the diagnostic tools to the regional epidemiology should strengthen malaria control in Mali. In particular, the use of RDTs that are capable of detecting pan-*Plasmodium* antigens, such as lactate dehydrogenase (pLDH) or aldolase, should be advocated in regions where non-*P. falciparum Plasmodium* species are endemic [14,49]. 

The main strength of our study was its country-wide wide coverage, which allowed us to obtain a comprehensive picture of RDTs performance and *Plasmodium* species distribution in Mali. In contrast, our study also has some limitations, the main one being potential selection bias. Indeed, the samples were collected through a healthcare-facility-based epidemiological surveillance system, and only 40% of the population attends these healthcare facilities. Another limitation of our study was the deliberate selection of sites due to political insecurity constraints. 

## 5. Conclusions

This study confirms that malaria rapid diagnostic tests (RDTs) contribute to significant improvement in the quality of malaria management in Mali. The results of this study suggest that the NMCP should recommend the use of RDTs with high analytical sensitivity that would be able to detect low parasitemia and set up a surveillance system for HRP2/3 deletion in patients, as it has already been documented in asymptomatic individuals in Mali. The NMCP should also recommend adapting RDTs’ specificity to the local *Plasmodium* species epidemiology to enhance malaria diagnosis in Mali.

## Figures and Tables

**Figure 1 ijerph-21-00228-f001:**
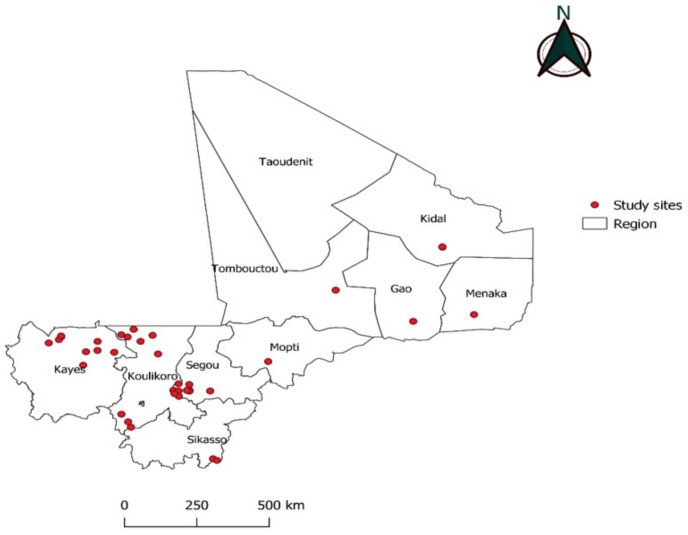
Location of the study sites in the nine administrative regions of Mali where malaria is endemic.

**Figure 2 ijerph-21-00228-f002:**
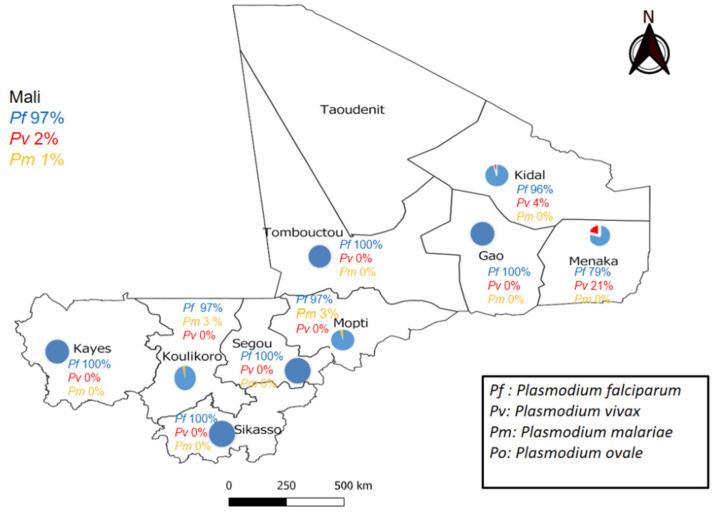
*Plasmodium* species distribution in the nine administrative regions of Mali where malaria is endemic.

**Figure 3 ijerph-21-00228-f003:**
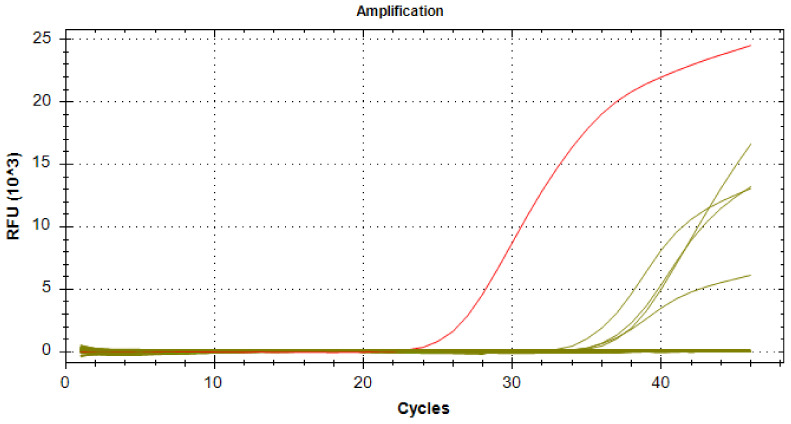
Amplification curves for real-time PCR targeting 18S rRNA of *Plasmodium vivax* species in Menaka (positive control in red and positive samples in green).

**Table 1 ijerph-21-00228-t001:** List of the primers and probes, targeting the *Plasmodium* spp. 18s rRNA genes, used in our study.

Target Species	Primers and Probes	Sequences
*Plasmodium* spp.	VAR ATS-F	CCCATACACAACCAAYTGGA
	VAR ATS-R	TTCGCACATATCTCTATGTCTATCT
	Var ATS-Probe	FAM-TRTTCCATAAATGGT
*Plasmodium falciparum*	Pf-F	TAGCATATATTAAAATTGTTGCAG
	Pf-R	GTTATTCCATGCTGTAGTATTCA
	Pf-probe	6FAM-CGGGTAGTCATGATTGAGTTCATTC
*Plasmodium malariae*	Pm-F	TAGCATATATTAAAATTGTTGCAG
	Pm-R	GTTATTCCATGCTGTAGTATTCA
	Pm-probe	6FAM- TGCATGGGAATTTTGTTACTTTGAGT
*Plasmodium ovale*	Po-F	TAGCATATATTAAAATTGTTGCAG
	Po-F R	GTTATTCCATGCTGTAGTATTCA
	Po-probe	6VIC- TGCATTCCTTATGCAAAATGTGTTC
*Plasmodium vivax*	Pv-F	AGCATATATTAAAATTGTTGCAG
	Pv-R	GTTATTCCATGCTGTAGTATTCA
	Pv-probe	6VIC- CGACTTTGTGCGCATTTTGC

Hybridization temperature was 60 °C for each PCR, and we used Taqman^TM^ hydrolysis probes [27].

**Table 2 ijerph-21-00228-t002:** Details of the total number collected and the proportions of positive and negative Malaria rapid diagnostic tests in each health center and administrative region.

Area	Health District	RDT Collection Site	Result		Total
			Positive N (%)	Negative N (%)	
Kayes	Diéma	Torodo	6 (2.2)	272 (97.8)	278
		Lakamané	3 (3.4)	89 (96.7)	92
		Lattakaf	0 (0.0)	47 (100)	47
		Débomassassi	0 (0.0)	48 (100)	48
		Koungo	3 (75.0)	1 (25.0)	4
		Lambidou	0 (0.0)	13 (100)	13
		Diancounté Camara	3 (6.4)	44 (93.6)	47
	Yélimané	Kodié	12 (19.0)	51 (81.0)	63
		Csréf	6 (4,1)	139 (95.9)	145
		Dogofry	0 (0.0)	23 (100.0)	23
		Bandiougoula	0 (0.0)	11 (100.0)	11
*sub total*			33 (4.3)	738 (95.7)	771
Koulikoro	Nara	Bagoini	0 (0.0)	50 (100.0)	50
		Mourdiah	5 (7.4)	68 (100.0)	73
		Kassakaré	4 (12.1)	29 (87.9)	33
		Alasso	7 (20.0)	28 (80.0)	35
		Tiapato	3 (14.3)	18 (85.7)	21
		Waourou	(0.0)	13 (100.0)	13
	Kangaba	Naréna	2 (1.6)	126 (98.4)	128
		Cscom Central	14 (28.0)	36 (72.0)	50
		Séléfougou	0 (0.0)	30 (100.0)	30
*sub total*			35 (8.1)	398 (91.9)	433
Sikasso	Kadiolo	Cscom central	110 (59.5)	75 (40.5)	185
		Kebeni	12 (16.4)	69 (85.2)	81
		Zégoua	28 (38.9)	44 (61.1)	72
*sub total*			150 (44.4)	188 (55.6)	338
Segou	Barouéli	Cscom Central	5 (41.7)	7 (58.3)	12
		Dioforogo	13 (65.0)	7 (35.0)	20
		Tamani	7 (35.0)	13 (65.0)	20
		NGara	16 (80.0)	4 (20.0)	20
		Tigui	20 (100.0)	0 (0.0)	20
		Bananido	7 (50.0)	7 (50.0)	14
		N’Gossola	5 (31.2)	11(68.8)	16
		Nianzana	20 (66.7)	10 (33.3)	30
		Yerebougou	16 (80.0)	4 (20.0)	20
		C.S. Réf.	8 (6.2)	122 (93.8)	130
		Ndjila	10 (50.0)	10 (50.0)	20
*sub total*			127 (39.4)	195 (60.6)	322
Mopti	Mopti	Soufouroulaye	79 (34.3)	151 (65.7)	230
		Fatoma	0 (0.0)	35 (100.0)	35
		Sévaré II	136 (91.9)	12 (8.1)	148
*sub total*			215 (52.1)	198 (47.9)	413
Tombouctou		Gourma Rharous	400 (66.9)	198 (33.1)	598
Gao	Gao	C.S. Réf. d’Ansongo	30 (13.4)	194 (86.6)	224
Menaka	Menaka	Menaka	21 (36.8)	36 (63.2)	57
Kidal	Kidal	Cscom d’Aliou	9 (14.3)	54 (85.7)	63
		C.S. Réf. de Kidal	20 (14.3)	120 (85.7)	140
*sub total*			29 (14.3)	174 (85.7)	203
Total			1019 (30.5)	2319 (69.5)	3338

**Table 3 ijerph-21-00228-t003:** Distribution of plasmodial species as identified using qPCR according to research sites.

Regions	*P. falciparum*		*P. vivax*		*P. malariae*		*P. ovale*		*P. falciparum + P. malariae*		*P. falciparum + P. vivax*		Total
	*n*	%	*n*	%	*N*	%	*n*	%	*N*	%	*n*	%	
Kayes	33	100	0	0	0	0	0	0	0	0	0	0	33
Koulikoro	28	96.6	0	0	1	3.4	0	0	1	3.4	0	0	29
Sikasso	30	100	0	0	0	0	0	0	0	0	0	0	30
Ségou	42	100	0	0	0	0	0	0	0	0	0	0	42
Mopti	33	97.1	0	0	1	2.9	0	0	0	0	0	0	34
Tombouctou	30	100	0	0	0	0	0	0	0	0	0	0	30
Gao	18	100	0	0	0	0	0	0	0	0	0	0	18
Kidal	21	91.3	1	4.3	0	0	0	0	0	0	1	4.3	23
Ménaka	15	78.9	4	21.1	0	0	0	0	0	0	0	0	19
Total	251	96.9	5	1.9	2	0.8	0	0	1	0.4	1	0.4	258

**Table 4 ijerph-21-00228-t004:** Diagnostic indices (95% confidence intervals), when compared to the qPCR gold standard, of the rapid diagnostic tests (RDT), compared to the qPCR gold standard, used for the biological diagnosis of malaria in each administrative region of Mali.

Regions	Sensitivity	Specificity	Youden	Kappa
Kayes	87.9% [76.9–90.7]	99.3% [96.9–100]	0.87 [0.74–0.91]	0.90 [0.77–0.94]
Koulikoro	96.6% [84.9–99.8]	98.7% [95.7–98.6]	0.95 [0.81–0.99]	0.94 [0.80–0.98]
Sikasso	80.0% [65.3–90.0]	96.0% [93.1–97.9]	0.76 [0.58–0.87]	0.76 [0.58–0.88]
Segou	71.4% [62.7–71.4]	100% [97.3–100]	0.71 [0.60–0.71]	0.79 [0.67–0.79]
Mopti	88.2% [79–88.0]	100% [98–100]	0.88 [0.76–0.88]	0.92 [0.80–0.92]
Tombouctou	93.3% [80.6–98.7]	97.3% [94.8–98.4]	0.91 [0.75–0.97]	0.88 [0.73–0.95]
Gao	94.4% [73.4–99.7]	92.6% [90.3–93.2]	0.87 [0.64–0.93]	0.68 [0.50–0.73]
Kidal	73.9% [54.5–87.8]	92.3% [89.4–94.4]	0.66 [0.44–0.82]	0.60 [0.40–0.74]
Ménaka	78.9% [59.0–92.0]	84.2% [74.2–90.7]	0.63 [0.33–0.83]	0.62 [0.32–0.80]

## Data Availability

The data that support the findings of this study are available from the corresponding author upon reasonable request.

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
