# Peer review of "Evaluation of the Performance of Rapid Diagnostic Tests for Malaria Diagnosis and Mapping of Different Plasmodium Species in Mali"

_ijerph, 2024, doi:10.3390/ijerph21020228_

Round 1

Reviewer 1 Report

Comments and Suggestions for Authors

Comments in file

Comments on the Quality of English Language

To the best of my knowledge, except for some minor corrections,  the manuscript is well written.

Author Response

Hi

Please see the attachment." in the box if you only upload an attachment.

 Thank you  very much.

Reviewer 2 Report

Comments and Suggestions for Authors

The study by Dembele et al aims at evaluating the performance of RDTs used in Mali by real time PCR and describing the distribution of Plasmodium species in the different administrative regions of Mali.

The distribution of different Plasmodium species, mainly falciparum, was consistent with previous studies.

The authors identify 39 false positive results out of 250 positive RDTs (15.6%) and 42 false negative results out of 1131 negative RTDs (3.7%). The highest false-negative rate (12/100) was found in the administrative region of Segou.

The work is clear and well written, even though with too many typos. English is fine.

Minor revisions

Line 27: “reach” should be modified in “reached”.

Line 37: a space should be introduced between 2020 and reference [1].

Line 41: the sentence “by the of the health system dysfunctions” needs to be revised.

Line 52: “vary” should be “varies” or “varied”.

Line 55: a space should be introduced between 2018 and reference [4].

Line 58: the acronym DHIS2 should be explained.

Line 60: “its” should be removed.

Line 87: the acronym “NMCPs” should be expanded. It is present later in the text (line 127), where it should instead be just NMCPs.

Line 90: the double space between “and” and “identification” should be removed. “They” should also be removed.

Line 94: there seems to be a double space between “high fatality” and reference [22].

Line 96: the full stop before references [23-35] should be removed.

Line 263: the full stop before reference [25] should be removed.

Line 266: niger should be Niger.

Line 267-68: Anopheles gambiae and Anopheles funestus should be in italics.

Line 288: “reamin” should be “remains”.

Lines 294-297: the authors state that “the sensitivity of RDTs was higher than 80 % in each region except in Kidal and Segou”. It is not clear whether it is due mainly to false positives, false negatives or both. This should be detailed. The authors then suggest that this result could be due the presence of species other than P. falciparum, but it seems that they only identify P. falciparum in these two administrations, except for one single case of P. vivax (Tab. 3). The sentence should be better explained.

Lines 299-302: the authors list a series of possible causes of false positives, but it seems that they also included some causes of false negatives (low parasite densities?). The sentence should be rephrased explaining which are the causes of false negatives and which of false positives.

307: a space is missing between “Mali” and references and space before full stop should be removed.

Author Response

Hi

Please see the attachment." in the box if you only upload an attachment.

 Thank you very much for yours comments.

Reviewer 3 Report

Comments and Suggestions for Authors

Dear Authors,

I found your paper interesting but at the same time found many errors and problems related to data and methods. A thorough review and revision is required to reorganize the available material in the paper for streamlining it . Facts and figures lack consistency through different sections which need to be looked at meticulously. I hope the detailed comments in the attached file will prove to be useful in improving the manuscript. 

Best Wishes !

Comments on the Quality of English Language

A thorough revision is required to improve the language and grammar for correctness and clarity. The intended meaning is not clear in many places due to language problems.

Author Response

Hi

Please see the attachment." in the box if you only upload an attachment.

Thank you very much for yours comments and all questions.

Reviewer 4 Report

Comments and Suggestions for Authors

Nevertheless this research discusses a significant public health issue but I believe it needs further clarity before it fulfills scientific soundness of the evidence it proclaims to have.

Since this study mentions at many places that RDTs used in the healthcare settings are only sensitive to Plasmodium Falciparum then why is there mention of "Malaria" as overall in the title as well as throughout the manuscript? Yes, this study found out other types of malaria through qPCR but that is a secondary finding, that is why primary objectives should be reflected in the title and focused in throughout the manuscript.

The Abstract, besides the abovementioned issues, describes vaguely results as well as conclusion.

Introduction part is not intelligently laid out. The statistics are used non-scientifically, e.g. number of malaria cases (line 48-9) without denominator does not signify anything concrete; In Lines 54-5 showing infant and child mortality from 6 to 59 months is an unstandardized way of presenting important data; Lines 57-8 are again pretty confusing for the readers 125% incidence? 81% target incidence? There are other irregularities on the dame pattern as well.

At some places, the abbreviations used (e.g. spp. TDR etc.) are also erratic and unstandardize and not have been listed at any place in the manuscript

There is no scientific explanation given for selection of samples from different geographical areas and final sampling followed no criteria. This is a serious bias for the current study.

The study should clearly state that sensitivity and specificity was measured for plasmodium falciparum type of malaria and finding other types of malaria were its significant by-product. The inclusion of other types of malaria to calculate sensitivity and specificity affects results accordingly.

Conclusion should be re-written and should not mimic results part. Conclusion will not be same as the title and objectives will be modified.

Comments on the Quality of English Language

There are very few grammatical errors but the overall expression to present the information scientifically is compromised. It needs a scientific writer to reflect true meanings of the work done. 

Author Response

Hi

Please see the attachment." in the box if you only upload an attachment.

 Thank you for yours all comments and questions.

Round 2

Reviewer 3 Report

Comments and Suggestions for Authors

ijerph-2756714-peer-review-comments-v1

Performance of rapid diagnostic tests for malaria diagnosis in  Mali

Pascal Dembélé et al.

The authors have made some minor revisions in the paper but all major concerns continue to exist in the revised manuscript also.

Abstract :

The suggested changes have not been addressed in the revised manuscript i.e., i) information about second objective pertaining to the distribution of various malaria parasites in the 9 administrative regions of Mali; ii)  inconsistency with the internationally accepted terminology e.g., biological testing, diagnostic indices, etc.; iii) correction of the language to make the scientific meaning clear for the readers using the internationally accepted terminology recommended by WHO.

Though the authors have changed the title of the manuscript, it does not address the above issues as the objectives and other things remain unchanged.

Introduction :

The authors have revised the introduction on the suggested lines to some extent and provided information on malaria prevalence for the year 2020 for 7 regions only against the suggestion of including the previous five years' data.

The author's cover letter is silent on the reasons why the suggested changes have not been made.

Methods :

Study Sites and Period of sample collection:

The suggested revisions have not been made and the confusion continues to exist which is not acceptable. The revised manuscript fails to provide clarity on the following important aspects:

i)                 Study sites

ii)               Study design

iii)             Study population 

iv)             Sampling technique

Sample collection /Sampling technique :  

It is not understandable why the authors refer the sample collection used in the study as epidemiological surveillance ? The authors have not used any ongoing systematic collection, analysis and interpretation of the data that can help in planning, implementation and evaluation of the public health practice used for malaria diagnosis in the nine administrative regions of Mali.

As per information provided by the authors all positive RDTs collected during the study period were not available to them. The total number of patients tested for malaria and positive by RDT during the study period has not been provided due to which it is not even possible to make out what %age of the total positive samples was tested by qPCR.

Strangely enough, the exclusion and inclusion criteria are not clear to the authors and the RDT invalid tests have been considered for exclusion. This criterion does not fit into the epidemiological criteria of inclusion/exclusion of samples and indicates a biased convenient sampling without following any scientific criteria to satisfy the objective of species distribution in the study area. Additionally, it does not conform to the statements of random selection given in the results section.

Case definitions  and Data analysis :

qPCR was the gold standard but only specificity was calculated using this test. For sensitivity only RDT was considered. In addition to this, the case definitions for true positive, false positive, true negative and false negative were also based on RDT results only.  The science and logic behind such calculations and case definitions is not understood and the methodology appears illogical as well as not conforming  to the accepted norms. Why wasn’t the gold standard test been used for the calculation of these indices?

Results :

The discrepancies and data mismatches continue to exist making this study questionable.

·        The actual number of sites in Table 2 is 39 only whereas in the text everywhere it is mentioned as 47. In Kayes and Koulikoro only 27 and 24 positive samples respectively were available(Table 2), then how were 30 samples selected and used for the qPCR; in Sikasso and Segou only 119 and 85 negative samples were available, then how were 150 samples used for qPCR? What was the qPCR result of the negative samples? Since it is not available in the tables and this paper, how was specificity calculated using the qPCR negative results?

·        The number of positive samples as per text under results sums up to 260 positive and 1236 negative samples amounting to a total of  1496  samples whereas the number mentioned in the text is 260 positive and 1397 negative. In the revised manuscript 6 invalid samples have also been included which is again controversial because as per the case definitions, the RDTs showing invalid tests (Missing C line) were already excluded from the study as per exclusion criterion.

·        How was the evaluation/performance of the RDTs calculated without considering the positive and negative results of the RDTs and PCR? Why have the corresponding results of the RDTs and PCR for the selected samples not been considered ? Please provide the reference followed for calculating sensitivity and specificity and designating True positives, false positives, and other case definitions.

S.

No.

Area

Table 2

(RDT results)

Table 3

(qPCR results)

As Per the Text under Results

+ve

-ve

+ve

+ve

-ve

1

Kayes

27

694

33

30

150

2

Koulikoro

24

398

28

30

150

3

Sikasso

138

119

30

30

150

4

Segou

127

85

42

30

150

5

Mopti

215

198

34

30

150

6

Tombouctou

400

198

31

30

150

7

Gao

30

194

18

30

150

8

Menka

21

36

19

21

36

9

Kedal

20

174

23

29

0

Total

1002

2096

258

260

1086

·        The number of samples tested does not match with those given in the abstract nor does the randomly selected number of a minimum of 30 positives and 150 negatives from each site match with Table 2. 

·        Check and correct the data provided in  the tables and interpretation in the text to resolve the discrepancies e.g., In table 3 it is shown that P. malariae was detected in 2 samples, 1 each from Koulikoro and Mopti and the %age has been calculated as 2.9%(1/34) and 3.57%(1/28)

These anomalies in data continue to exist in the revised manuscript also and again point to biased sample selection and data compilation methods.

·        Referring to Table 4 the authors mention  81 discordant RDT results, namely 15.6 % (39/250) false-positive and 3.7% (42/1131) false-negative results had discordant cases. Where is it possible to see this in Table 4? The meaning of ‘had discordant cases’ in the above sentence is not clear. Particularly high false-negative rate (12/100) in Segou is also not clear from this table.

·        How do the authors expect the numbers and results written in the text to be correlated with Table 4? The Table 4 heading reads as ‘Diagnostic indices [95% confidence intervals] when compared to the qPCR gold standard, of the rapid diagnostic tests (RDT) used for the biological diagnosis of malaria in each administrative region of Mali’. Sensitivity and specificity are mentioned in the table but as per the case definitions Sensitivity was not calculated using RDTs and qPCR.

Discussion and Conclusion :

Not considered adequate and satisfactorily revised given the above existing anomalies and discrepancies.

Comments on the Quality of English Language

English language needs improvement especially in the new text added to the manuscript. 

Author Response

Hello ,We are delighted to send you the answers to your questions, comments and suggestions, which have enabled us to correct many errors in the manuscript. First of all, we'd like to apologize and congratulate you on your valuable contribution. Thank you for your tremendous efforts.
Sincerely
We wish you a happy reception.

Reviewer 4 Report

Comments and Suggestions for Authors

The scientific language still needs improvement at many places. e.g.,

Lines (47-48) "In Mali, malaria is the primary reason for consultations,...." What is meant by consultations here??

Lines (51-52) phrases like "most lethal" lethal etc. should not be used

Lines (98-99)  47 compared 47 sites????? "i) the performance of rapid 98 diagnostic tests collected from 47 compared 47 sites in 9 administrative regions with qPCR..."

AND

Line (103) Sub-heading Study Design does not reflect any information on study's design rather discusses Study area

Line (115) "Mali's population is characterized by its extreme youth." What does this statement signify?

Line (124-5) The region of Taoudenit was not surveyed because it is not endemic for malaria due to its Saharan climate. Better cite this statement appropriately

Line (320) Limit? Better write complete word

Lines (322-5) "...with only 40% of patients consulting the facilities, which does not give an exhaustive list of patients. Another limitation of the study was the reasoned choice of our sites for reasons of insecurity. But this study has the advantage to cover the whole country to give an idea of the mapping of the performance of RDTs and parasitic species in Mali."

These sentences are in-coherent..... Needs to be revised

Conclusion also needs to be elaborative

Comments on the Quality of English Language

English language still needs to be revised scientifically

Author Response

(The authors gave the same response as above.)
